## [Reviewer comments · BMJ Open]

ARTICLE DETAILS

TITLE (PROVISIONAL)	Did previous involvement in research affect recruitment of young people with cerebral palsy to a longitudinal study of transitional health care?
AUTHORS	Garcia Jalon, Elena Guiomar Merrick, Hanna; Colver, Allan; Linden, Mark

VERSION 1 – REVIEW

REVIEWER	Marloes van Gorp Netherlands
REVIEW RETURNED	05-Dec-2019

GENERAL COMMENTS	This is a decent study design and report, however, I am somewhat confused by the two parts of statistical testing and think the consistency within the paper can be improved. See my comments for each section outlined below Abstract - The objective (to assess the effect of previous research on recruitment) does not entirely match the statistics described here (to assess predictors of recruitment, including previous research AND demographic and CP characteristics) Strengths and limitations - Phrasing of second bullet is awkward Introduction - The sentence before the research aim suggests that studying differences between direct and indirect contact is also an aim, but this is not mentioned in the aim, which is slightly confusing Methods - data analysis: it is not fully clear to me why you would test differences in responder/refuser subgroups as well as study predictors for response in logistic regression. Please make this more clear, or perhaps only do the regression to make the paper more straightforward. - data analysis: the logistic regression is described to study predictors and control for characteristics here, however in the results and interpretation all variables are treated the same (and in the abstract all are mentioned as possible predictors). Please be consistent throughout in your aim and use fitting analyses: are you studying predictors (from all variables) or studying the effect of previous recruitment/contact for studies (and where does the direct/indirect approach variable fit)? Also, please elaborate slightly on the model: is it a multivariate model including all mentioned variables, or is a stepwise approach for prediction modelling used
--

	(I guess the first is true, but please make that clear here)? Results  - This is my main problem: The results of the subgroup (refuser/responder) and predictor analyses seem to be in opposite directions. Is this correct? Then please discuss, because this is very confusing: 1. older age predicts lower odds of refusal (table 2), but the mean age in the recruitment group is lower (table 1). 2. Previous recruitment to studies predicts higher odds of refusal (table 2), but the proportions of individuals recruited to previous studies are higher in the 'recruited' group (table 1 and in text: 53% of recruited had previously taken part in a study). - Table 2: It seems appropriate to categorize GMFCS in fewer categories, because of low numbers Discussion  - second paragraph: again this seems contradictory: those recruited were younger on average (first sentence), but older individuals were less likely to refuse (3rd sentence) - second paragraph, last sentence: besides a borderline p-value, the difference 0.3 yrs also seems small, maybe discuss the relevance? References The majority of your references is somewhat dated.
--	--

REVIEWER	Gunvor Lilleholt Klevberg Oslo University Hospital, Norway
REVIEW RETURNED	31-Jan-2020

GENERAL COMMENTS	Summary of the review: Thank you for the opportunity to review this well-written and well-structured paper regarding recruitment of young people with CP in research. According to the abstract, the objective of the study was “to assess whether being contacted about participating in previous research affected recruitment to a Transition study from child to adult healthcare services of young people with CP”. The main findings are that older youth with CP, or youth who had previously been approached about research participation, were more likely to be recruited to the study. However, youths who had in fact participated in previous research were more likely to refuse to participate in the study. There was no difference in the likelihood of recruitment to the study between a direct and an indirect recruitment approach. The authors conclude that the recruitment approach did not affect recruitment, yet that there may be evidence of research fatigue due to the fact that participants who had previously been recruited to a study were less likely to accept a new invitation. The authors communicate an important message to researchers who utilize registries as a recruitment platform for research studies, and remind us to be cautious and consider the potential risk of research fatigue for young people with CP who are included in such registries. I am not a native English speaker, but in my opinion the paper is easy to read and excellently written.
--

Below are some reflections on the content and structure of the paper.

Suggestions for revisions are given with numbers in parentheses prior to the sentence.

Title: The title successfully reflects the content of the paper

Abstract: The abstract successfully covers the content of the paper, by stating the objective, describing the main methods, findings and interpretations of these.

Strength and limitations: The bullet-points highlight the most important point to address

Introduction and aims: The background information is brief, yet covers sufficient information to introduce the reader to the topic of the paper.

- (1) The research questions or study objective should be stated more clearly in the introduction section.
- (2) P.3, line 21: Sentence begins with "This study[...]". I assume "This study" refers to the Transition Research Programme, yet this needs to be explicitly stated. Furthermore, if "This study" is the same as "Transition study" which is a term used several places in the manuscript, this should also be clarified.
- (3) P.3, line 28: "Differences in the operating procedures of the two registers also allowed.....". What does "also" refer to in this sentence? I cannot see that the authors have described any other differences between the two registers than the direct/indirect approach. Are there other operating procedures that are different and which should be addressed?

Methods

The methods are well written, and the definition of terms are very useful.

- (4) P.4, line 8/9: Here is "the Transition study" used for the first time. Please describe earlier whether this refers to "the Transition Research Programme" or a different study.
- (5) P.4, line 31: You describe age 14-18 years as inclusion criteria for the study. Although there are different approach to consent to the two registers, I would like you to include a sentence in the method section regarding self-consent or parent consents – since this is also a point addressed in the Discussion.
- (6) P. 4, line 58: To my understanding, the GMFCS classifies the motor functioning of a person rather than motor impairments.

Results: The results are described in adequate detail, documenting the significant findings, are easy to read and logically presented.

Tables and figures: The tables are tidy and clear, and represent useful illustrations of recruitment procedure and main results.

Discussion: The discussion appropriately reflects the findings and is logically structured.

- (7) P. 10, line 8/9: "The majority who declined stated they were not interested in the study". In my opinion, this is an important statement that points to the importance of user involvement

	already when planning a study and defining research questions. I feel this statement deserves a little more attention. (8) P. 10, line 23-38. Your findings are somewhat confusing. At the one hand, participants in the “Recruited” group were younger than in the “Refusers” group. On the other hand, results from your logistic regression show that “Older individuals were less likely to refuse” to participate. Please present an explanation to these seemingly contradicting results. (9) P. 10, line 49. “[.] in the Transition study the focus was on young people giving consent”. Please provide information on the consent procedure in the Methods section. (10) P. 11, line 40/41: The point regarding “subset of young people with CP who are not interested in research” goes together with my comment number 7 and should be further addressed in context of user involvement in research planning. Conclusion: The conclusion points towards the most important findings, and is supported by the results. Along with a growing number of national registries, your indications of “research fatigue” and the importance of user involvement in research are important point so be communicated to researchers involved in registries.
--	--

VERSION 1 – AUTHOR RESPONSE

Reviewer 1 Comments to Author and Author’s responses:

Abstract

- The objective (to assess the effect of previous research on recruitment) does not entirely match the statistics described here (to assess predictors of recruitment, including previous research AND demographic and CP characteristics)

Response: Logistic regression assessed whether Recruited vs. Refusers was predicted by the number of previous studies approached about or taken part in and/or method of approach (Direct vs. Indirect) controlling for demographic characteristics (age when first approached, sex, home location) and CP characteristics (subtype, GMFCS and intellectual impairment). The main factors the authors were interested on whether they had an effect of recruitment or not were previous involvement on research and method of approach. However, the role that demographic and CP characteristics could have could not be ignored and thus authors controlled for these factors by including them in the analysis.

*The **Objective** and **Design and Method** sections within the abstract has been modified to clarify the point raised by the reviewer. Please refer to revised version of the manuscript copy version with tracked changes, page 24 lines 7, 18 and 19.*

Strengths and limitations

- Phrasing of second bullet is awkward

*Response: phrasing has been reviewed and amended. Please see reviewed manuscript under the section **Strengths and limitations** second bullet point which now reads “The study of the potential effect of prior involvement in research on recruitment was novel”. We have also rephrased bullets four*

and five. Please refer to revised version of the manuscript copy version with tracked changes, page24 lines 52 and page 25 lines 5 and 10.

Introduction

- The sentence before the research aim suggests that studying differences between direct and indirect contact is also an aim, but this is not mentioned in the aim, which is slightly confusing

Response: the authors have amended the goal of the study in the introduction section to mirror that of the abstract. Please refer to revised version of the manuscript copy version with tracked changes, page25 lines 25 to 27.

Methods

- data analysis: it is not fully clear to me why you would test differences in responder/refuser subgroups as well as study predictors for response in logistic regression. Please make this more clear, or perhaps only do the regression to make the paper more straightforward.

Response: Responding by declining to take part in research (active refusal) and not responding to invitations to research (passive refusal) are different. There is the potential that logistic regression test in this study should include a dependent variable with three categories, this is "Recruited", "Refusers-active refusal" and "Refusers-passive refusal". The authors consider this option rather than to have a dependent variable with two categories as the manuscript reports, i.e. "Recruited" and "Refusers".

However, this would have made each group too small and thus compromised the analysis. Therefore, it was deemed necessary to analyse whether there were differences between the sub-groups within Refusers, i.e. Non-responders (passive refusal) and Responders who declined (active refusal). As there were not differences, it was deemed appropriate to conduct the logistic regression analysis with a dependent variable that had two categories as explained above. The lack of differences between these subgroups is a relevant point worth to include and discuss in the manuscript. It is also reported for transparency reasons and to offer a full picture to readers as to how the analyses were conducted.

- data analysis: the logistic regression is described to study predictors and control for characteristics here, however in the results and interpretation all variables are treated the same (and in the abstract all are mentioned as possible predictors). Please be consistent throughout in your aim and use fitting analyses: are you studying predictors (from all variables) or studying the effect of previous recruitment/contact for studies (and where does the direct/indirect approach variable fit)? Also, please elaborate slightly on the model: is it a multivariate model including all mentioned variables, or is it a stepwise approach for prediction modelling used (I guess the first is true, but please make that clear here)?

Response: amendment applied to abstract to reflect on reviewer's comment stating that "Logistic regression was used to assess contact about and recruitment to previous research and method of approach as predictors of recruitment, controlling for demographic and CP characteristics." Please refer to revised version of the manuscript copy version with tracked changes, page 24 lines 7, 18 and 19.

The method used to conduct the logistic regression was a standard enter method, that is all the variables were entered simultaneously and the predictive power of the combination of all of the variables was considered. Given the small sample size in this study, a stepwise method was not considered appropriate as the variables used could be prone to overestimate the success of the predictors. This has been clarified in the "Data analysis" section of the manuscript. Please refer to revised version of the manuscript copy version with tracked changes, page 27 line 54.

Results

- This is my main problem: The results of the subgroup (refuser/responder) and predictor analyses seem to be in opposite directions. Is this correct? Then please discuss, because this is very confusing: 1. older age predicts lower odds of refusal (table 2), but the mean age in the recruitment group is lower (table 1). 2. Previous recruitment to studies predicts higher odds of refusal (table 2), but the proportions of individuals recruited to previous studies are higher in the 'recruited' group (table 1 and in text: 53% of recruited had previously taken part in a study).

Response: Differences in age between Recruited and Refusers is marginal and the difference is only just significant with a p value of 0.05. Other considerations are the narrow age range of the sample and the non-normal distribution of the data for the full sample with a p value in the Shapiro Wilk test of 0.000. When looking at the distribution of the data regarding age within Recruited and Refusers, neither data set was normally distributed with Recruited showing positive skewness (i.e. towards younger age) and Refusers showing a negative skewness (i.e. towards older age). A document with supplementary data showing the results of the normality test for the variable of age in the overall sample and each the Recruited and Refusers groups has been submitted with the amended document.

Regarding the proportions of individuals recruited to previous studies, while the comments of the reviewer are correct, we also need to take into consideration the proportions within each group, Recruited and Refusers. While within the Recruited group there is a more homogenous distribution of participants who took part in 1, 2 or 3 and more studies, within Refusers, there are larger differences between those who had taken part in 1, 2 or 3 and more studies. This will have an effect on logistic regression output.

- Table 2: It seems appropriate to categorize GMFCS in fewer categories, because of low numbers

Response: distribution on GMFCS was not representative of the overall CP population given the selection criteria for the Transition study. Thus, it was decided not to further manipulate this variable by using fewer categories.

Discussion

- second paragraph: again this seems contradictory: those recruited were younger on average (first sentence), but older individuals were less likely to refuse (3rd sentence)

*Response: Reasons for the apparent contradictory results regarding age have been discussed above in reviewer's first point regarding the **Results** section.*

Differences in mean age between Recruited and Refusers is marginal (standard deviations are also only slightly different between the two groups) and the difference is only just significant with a p value of 0.05. Equally, the age range in both groups is narrow and data was not normally distributed. This is why as indicated in the discussion section, results from the logistic regression analysis should be considered with caution. Please refer to revised version of the manuscript copy version with tracked changes, page 33 lines 12 to 17 to see amendment text to address this point.

- second paragraph, last sentence: besides a borderline p-value, the difference 0.3 yrs also seems small, maybe discuss the relevance?

*Response: this sentence in the **Discussion** section of the revised manuscript has been amended to include the difference in age between Recruited and Refusers is also small (0.3 years). Please refer to revised version of the manuscript copy version with tracked changes, page 33 line 15.*

References

The majority of your references is somewhat dated.

Response: This study is assessing recruitment to a longitudinal Transition study and thus references used in this manuscript are relevant to the recruitment methods used in the Transition study and which address issues specific to the Transition study and its context.

*The main author has looked at the references used in each of the sections of the manuscript especially the **Introduction** section and the **Discussion**. In the **Introduction** references are included to support authors' statement regarding difficulties surrounding the recruitment of young people into research and how this limits the quality of the research. More recent publications continue to refer to these historical issues. The authors have included two of these references in the revised manuscript (references 4 and 13). Other references included in the **Introduction** are used to support statements about the Transition study and the disease specific registers used to identify and recruit young people to the Transition study. Please refer to revised version of the manuscript copy version with tracked changes, page 37 lines 36 to 40 and page 38 lines 23 to 27 to see additional references.*

*The references included in the **Discussion** are relevant to specific aspects of the recruitment strategies used in the Transition study. More recent publications explore methods of recruitment different from those used in the Transition study. Some of these publications have been included as references in the revised manuscript to address aspects of a query raised by reviewer 2 (references 4 and 27). Please note these publications were published after recruitment to the Transition study was completed, and while their results can inform the revised manuscript in hindsight, it is important to discuss our results within a context relevant to the Transition study. Please refer to revised version of the manuscript copy version with tracked changes, page 37 line 36 to 40 and page 39 lines 32 to 34 to see additional references.*

Reviewer 2 Comments to Author and Author's responses:

Conclusion of review: Accept with minor revisions

Summary of the review:

Thank you for the opportunity to review this well-written and well-structured paper regarding recruitment of young people with CP in research.

According to the abstract, the objective of the study was "to assess whether being contacted about participating in previous research affected recruitment to a Transition study from child to adult healthcare services of young people with CP".

The main findings are that older youth with CP, or youth who had previously been approached about research participation, were more likely to be recruited to the study. However, youths who had in fact participated in previous research were more likely to refuse to participate in the study. There was no difference in the likelihood of recruitment to the study between a direct and an indirect recruitment approach.

The authors conclude that the recruitment approach did not affect recruitment, yet that there may be evidence of research fatigue due to the fact that participants who had previously been recruited to a study were less likely to accept a new invitation.

The authors communicate an important message to researchers who utilize registries as a recruitment platform for research studies, and remind us to be cautious and consider the potential risk of research fatigue for young people with CP who are included in such registries.

I am not a native English speaker, but in my opinion the paper is easy to read and excellently written.

Below are some reflections on the content and structure of the paper.

Suggestions for revisions are given with numbers in parentheses prior to the sentence.

Title: The title successfully reflects the content of the paper

Response: Thank you for the very positive comments. However, following instructions by the Editor, the title has been modified. Please refer to revised version of the manuscript copy version with tracked changes, page 23 lines 3 and 4.

Abstract: The abstract successfully covers the content of the paper, by stating the objective, describing the main methods, findings and interpretations of these.

*Response: Thank you for the very positive comments. However, amendments have been applied to clarify questions raised by the other reviewer. The **Objective** and **Design and Method** sections within the abstract has been modified to clarify these queries. Please refer to revised version of the manuscript copy version with tracked changes, page 24 lines 7, 18 and 19.*

Strength and limitations: The bullet-points highlight the most important point to address **Introduction and aims:** The background information is brief, yet covers sufficient information to introduce the reader to the topic of the paper.

(1) The research questions or study objective should be stated more clearly in the introduction section.

*Response: the last paragraph in the **Introduction** section has been modified to address this point and clearly states what is the goal of the study. Please refer to revised version of the manuscript copy version with tracked changes, page 25 lines 45 to 49.*

(2) P.3, line 21: Sentence begins with "This study[...]". I assume "This study" refers to the Transition Research Programme, yet this needs to be explicitly stated. Furthermore, if "This study" is the same as "Transition study" which is a term used several places in the manuscript, this should also be clarified.

*Response: "This study" in P3 line 21 of the original manuscript submitted for review refers to the Transition longitudinal study. To clarify this point the authors have amended such line in the introduction by adding the word "longitudinal" and have included a definition for the Transition study under **Study design and participants** subsection within the **Methods** section. Please refer to revised version of the manuscript copy version with tracked changes, page 26 lines 24 to 28.*

(3) P.3, line 28: "Differences in the operating procedures of the two registers also allowed.....". What does "also" refer to in this sentence? I cannot see that the authors have described any other differences between the two registers than the direct/indirect approach. Are there other operating procedures that are different and which should be addressed?

Response: With the wording "also" the authors had tried to communicate that as the longitudinal study had offered the opportunity to evaluate the potential differences in recruitment of young people with CP using two population-based UK registers for which participation in previous research was known, differences in the operating procedures of such population-based registers also allowed the assessment of the effect of direct and indirect approach methods.

The authors had have rephrase this sentence to prevent any confusion. The above sentence now reads as "Equally, differences in the operating procedures of the two registers allowed". Please refer to revised version of the manuscript copy version with tracked changes, page 25 line 38.

Methods

The methods are well written, and the definition of terms are very useful.

(4) P.4, line 8/9: Here is “the Transition study” used for the first time. Please describe earlier whether this refers to “the Transition Research Programme” or a different study.

*Response: The authors have included a definition for the Transition study under **Study design and participants** subsection within the **Methods** section. Please refer to revised version of the manuscript copy version with tracked changes, page26 lines 24 to 28.*

(5) P.4, line 31: You describe age 14-18 years as inclusion criteria for the study. Although there are different approach to consent to the two registers, I would like you to include a sentence in the method section regarding self-consent or parent consents – since this is also a point addressed in the Discussion.

Response: Information regarding consent to the Transition study has been included in this paragraph. Please refer to revised version of the manuscript copy version with tracked changes page 27 lines 5 to 9.

(6) P. 4, line 58: To my understanding, the GMFCS classifies the motor functioning of a person rather than motor impairments.

Response: this has been corrected and “impairment” has been substituted for “function”. Please refer to revised version of the manuscript copy version with tracked changes, page 27 line 31.

Results: The results are described in adequate detail, documenting the significant findings, are easy to read and logically presented.

Tables and figures: The tables are tidy and clear, and represent useful illustrations of recruitment procedure and main results.

Discussion: The discussion appropriately reflects the findings and is logically structured.

(7) P. 10, line 8/9: “The majority who declined stated they were not interested in the study”. In my opinion, this is an important statement that points to the importance of user involvement already when planning a study and defining research questions. I feel this statement deserves a little more attention.

*Response: The issue of user involvement has been included in the discussion as advised by the reviewer. This issue has been discussed after the second paragraph and in the last paragraph of the **Discussion** section. The main aspects discussed regarding user involvement (referred to as patient and public involvement – PPI - in the revised manuscript) are that although there is evidence pointing at the positive impact of PPI in recruitment, this is limited and more robust research is necessary. Despite PPI, some young people may still refuse to take part in research. Nevertheless, researchers should explore/consider all possible recruitment strategies, report these as well as PPI. Please refer to revised version of the manuscript copy version with tracked changes, page 32 lines 24 to 54.*

(8) P. 10, line 23-38. Your findings are somewhat confusing. At the one hand, participants in the “Recruited” group were younger than in the “Refusers” group. On the other hand, results from your logistic regression show that “Older individuals were **less** likely to refuse” to participate. Please present an explanation to these seemingly contradicting results.

*Response: Differences in age between Recruited and Refusers is marginal and the difference is only just significant with a p value of 0.05. Other considerations are the narrow age range of the sample and the not normal distribution of the data for the full sample with a p value in the Shapiro Wilk test of 0.000. When looking at the distribution of the data regarding age within Recruited and Refusers, neither data set was normally distributed with Recruited showing positive skewness (i.e. towards younger age) and Refusers showing a negative skewness (i.e. towards older age). This has been addressed in paragraph 5 of the **Discussion** section. Please refer to revised version of the*

manuscript copy version with tracked changes, page 33 lines 12 to 17. A document with supplementary data showing the results of the normality test for the variable of age in the overall sample and each the Recruited and Refusers groups has been submitted with the amended document.

(9) P. 10, line 49. “[..] in the Transition study the focus was on young people giving consent”. Please provide information on the consent procedure in the Methods section.

Response: Information regarding consent to the Transition study has been included in the Methods section. Please refer to revised version of the manuscript copy version with tracked changes, 27 lines 5 to 9.

(10) P. 11, line 40/41: The point regarding “subset of young people with CP who are not interested in research” goes together with my comment number 7 and should be further addressed in context of user involvement in research planning.

*Response: This point has been addressed paragraphs 3 and 4 and the last of the **Discussion** section (revised version). Please refer to revised version of the manuscript copy version with tracked changes, page 34 lines 43 to 46.*

Conclusion: The conclusion points towards the most important findings, and is supported by the results. Along with a growing number of national registries, your indications of “research fatigue” and the importance of user involvement in research are important point so be communicated to researchers involved in registries.